# Chemical Review of Gorgostane-Type Steroids Isolated from Marine Organisms and Their ^13^C-NMR Spectroscopic Data Characteristics

**DOI:** 10.3390/md20020139

**Published:** 2022-02-14

**Authors:** Fahd M. Abdelkarem, Mohamed E. Abouelela, Mohamed R. Kamel, Alaa M. Nafady, Ahmed E. Allam, Iman A. M. Abdel-Rahman, Ahmad Almatroudi, Faris Alrumaihi, Khaled S. Allemailem, Hamdy K. Assaf

**Affiliations:** 1Department of Pharmacognosy, Faculty of Pharmacy, Al-Azhar University, Assiut 71524, Egypt; dr.fahd@azhar.edu.eg (F.M.A.); m_abouelela@azhar.edu.eg (M.E.A.); alaanafady@azhar.edu.eg (A.M.N.); ahmedallam@azhar.edu.eg (A.E.A.); hamdyss200@azhar.edu.eg (H.K.A.); 2Department of Pharmacognosy, Faculty of Pharmacy, South Valley University, Qena 83523, Egypt; emanabdelraheem@svu.edu.eg; 3Department of Medical Laboratories, College of Applied Medical Sciences, Qassim University, Buraydah 51452, Saudi Arabia; aamtrody@qu.edu.sa (A.A.); k.allemailem@qu.edu.sa (K.S.A.)

**Keywords:** gorgostane, marine organisms, ^13^C-NMR, steroids

## Abstract

Gorgostane steroids are isolated from marine organisms and consist of 30 carbon atoms with a characteristic cyclopropane moiety. From the pioneering results to the end of 2021, isolation, biosynthesis, and structural elucidation using ^13^C-NMR will be used. Overall, 75 compounds are categorized into five major groups: gorgost-5-ene, 5,6-epoxygorgostane, 5,6-dihydroxygorgostane, 9,11-secogorgostane, and 23-demethylgorgostane, in addition to miscellaneous gorgostane. The structural diversity, selectivity for marine organisms, and biological effects of gorgostane steroids have generated considerable interest in the field of drug discovery research.

## 1. Introduction

Marine natural products are an untapped reservoir for discovering biologically active phytomolecules, with a significant array of bioactivities reported against many aliments [1,2]. Antibiotic, antifungal, anti-viral, cytotoxic, and anti-tumor activities are among the reported pharmacological effects of marine-derived molecules [3,4]. Marine organisms have been considered an extraordinarily rich source of new sterols with a core ring system, side chains, and unusual oxygenation patterns on their A–D rings [5,6,7,8]. The origin of these marine invertebrates’ sterols is complicated because they may be formed from dietary origins or synthesized by a symbiont and later modified biochemically in the invertebrate [9]. Marine organisms have yielded many sterol metabolites with unusual side chains [10].

Gorgosterol was the first sterol reported to contain a cyclopropane ring on the side chain, and since then, several additional sterols with this ring structure and sterols with polyoxygenated functionalities have been isolated [10]. The reported gorgostane-type steroids from marine organisms possess certain biological activities, including anti-inflammatory activity reported in stoloniferone S isolated from soft coral *Clavularia viridis* [11], antibacterial and antifungal activities in the in vitro bioassay reported in 11α-acetoxy-gorgostane-3β,5α,6β,12α-tetraol and 12α-acetoxy-gorgostane-3β,5α,6β,11α-tetraol isolated from soft coral *Sarcophyton* species [12], and cytotoxic activity reported in klyflaccisteroids C-F isolated from soft coral *Klyxum flaccidum* [13].

Gorgostane-type steroids have some structural variability in the number of carbon atoms and the degree of oxygenation patterns. According to the number of carbon atoms and structural features, gorgostane-type steroids can be classified into five groups. In this review, 75 gorgostane-type steroids with their ^13^C-NMR spectroscopic data are summarized. Hopefully, this review will contribute to the elucidation of the structure and identification of this class of compounds.

## 2. Distribution of Gorgosteroids among Marine Organisms

Several classes of sterols were isolated from different marine organisms [14]. However, gorgostane is a widely occurring group of sterols in marine organisms and is isolated mainly from the order Alcyonacea (soft coral). The parent compound, gorgosterol, was isolated for the first time from the octocoral *Plexaura flexuosa* [15]. Seventy-five compounds were isolated, seventy-three from soft corals and two compounds from the algal-bearing gorgonian coral *Pinnigorgia* sp. (Table 1). The significant gorgostane steroids are 16 compounds isolated from the soft coral *Isis hippuris*, 11 compounds isolated from the genus *Sinularia*, and 9 compounds from *Klyxum flaccidum*. The four reported epidioxygorgostane steroids, 5α,8α-epidioxy-23,24-didemethylgorgost-6-ene-3β-ol, 5α,8α-epidioxy-23-demethylgorgosta- 6,9(11)-dien-3β-ol, 5α,8α-epidioxygorgost-6-en-3β-ol, and 5α,8α-epidioxygorgost-6,9(11)-dien-3β-ol, were isolated only from the genus *Sinularia*. Nine of the ten compounds, which belong to 5, 6-epoxy gorgostane, were isolated from *Isis hippuris*. Also, the majority of demethylgorgostane compounds were separated from *Clavularia viridis*.

## 3. Biosynthesis of Gorgostane Steroids

Sterols with side chains containing the 22,23-cyclopropane group have been encountered only in soft corals that live symbiotically with zooxanthellae. Several experiments were performed to explain the biosynthetic pathways of gorgosterols [52] using the cell-free extracts of the dinoflagellates *Peridinium foliaceum*, *Crypthecodinium cohnii,* and the cultured zooxanthella symbiont of *Cassiopea xamachana*. They found that a decrease in S-adenosylmethionine concentration concomitantly with an increase in dimethylpropiothetin biosynthesis is linked with the attenuation of the production of gorgosterol in aposymbiotic zooxanthellae [53]. In another study, the pseudoplexaurids *Pseudoplexaura porosa, P. flagellosa,* and *P. wagenaari* and *Pseudopterogorgia americana* soft corals were used as a source of zooxanthellae to detect the conversion of labelled farnesyl pyrophosphate to squalene. They concluded that zooxanthellae obtained from *P. porosa* contributed to the part of the pathway from mevalonate to gorgosterol (Figure 1) that encloses the formation of squalene [54].

## 4. Gorgostane Steroids

Gorgostane is a steroid with a basic molecular formula of C_30_ H_52_ (Figure 2). Most are pentacyclic, with the exception of secogorgosterols, which are tetracyclic steroids. Both gorgosterol and secogorgosterol have a side chain containing a characteristic cyclopropane moiety. The cyclopropane moiety of the gorgostane steroid (C-22, C-23, and C-30) showed characteristic signals in ^13^C NMR at 32.1, 25.8, and 21.3 ppm, respectively. According to the number of rings, the presence and position of unsaturation, and the number of substituents, gorgostane can be classified into different groups.

### 4.1. Gorgost-5-ene

Gorgost-5-ene is characterized by the presence of unsaturation between C-5 and C-6 (Figure 3 and Table 2, Table 3 and Table 4). In ^13^C-NMR, the two olefinic carbons, C-5 and C-6, showed characteristic signals at 140.8 and 121.7 ppm, respectively. The previous values are affected by the degree of substitution of the A and B rings. The substitution of the core steroid ring system was mainly hydroxyl, carbonyl, and acetoxy groups. This substitution led to an upfield or downfield ^13^C-NMR chemical shift of the substituted carbon and those nearby [6]. Additional double bonds were located at C-9, C-11 and C-25, and C-26, as in compounds klyflaccisteroid E (**5**) and gorgosta-5,25-dien-3β-ol (**22**), with characteristic ^13^C-NMR signals at 149.0 (C-11), 121.1 (C-9) and 156.9 (C-25), and 106.0 (C-26) [13,25]. The parent compound of this steroid is gorgost-5-ene, and the substitution usually occurs at C-1, C-7, C-11, C-12, and C-18. Isolation of gorgost-5-ene with a hydroxyl group at C-7 leads to a high downfield shift in the ^13^C-NMR of C-7, as in crassumsterol (**2**) and klyflaccisteroid E (**5**), to 65.3 and 74.9 ppm, respectively [13,17]. The difference in the chemical shift was due to the *α*-orientation of the hydroxyl group in crassumsterol and *β*-orientation in klyflaccisteroid [13,17]. Gorgost-5-ene steroids were isolated with a ketonic carbonyl group only at position 11, as in klyflaccisteroid C (**7**), and at position 7 and 11, as in klyflaccisteroid D (**8**), with a characteristic ^13^C-NMR signal at 214.1 (C-11) and 211.6 (C-11) in klyflaccisteroid D [13]. Another common substitution is the acetate group, located at C-3, C-11, C-12, C-15, and C-18 with a characteristic signal at 170.0 to 173.4 ppm [22,23] and 3*β*-Acetoxy-1α,11α-dihydroxygorgost-5-en-18-oic acid (**18**) with a characteristic carboxylic group at position 18 with a signal at 176.9 ppm. The existence of a carboxylic group at position 18 leads to a downfield shift in C-13 by 3 ppm in comparison with gorgost-5-en-3*β*-ol (**1**) [16,20]. 

### 4.2. 5,6-Epoxygorgostane

5,6-Epoxygorgostane is characterized by the presence of an epoxide of β orientation at C-5 and C-6 (Figure 4 and Table 5). In ^13^C-NMR, the two carbons of the epoxide, C-5 and C-6, showed characteristic signals at 62.9 and 63.7 ppm, respectively. The degree of substitution of the A and B rings greatly affects the previous chemical shifts of C-5 and C-6. The substitutions, mainly hydroxyl or acetoxy groups, usually occur at C-1, C-3, C-7, C-11, C-12, and C-15. 5β,6β-Epoxygorgosterol (**23**) [26] is the parent compound of this group, which has only one hydroxyl group at C-3. 5β,6β-Epoxygorgostane-1α,3β,11α,12β-tetrol (**24**) [23] has a trihydroxy substitution at C-1, C-11, and C-12 with chemical shifts 73.8, 73.8, and 83.6 ppm, respectively. 5β,6β-Epoxygorgostane-3β,11α,12β-triol 12-acetate (**25**) [22] has a hydroxyl group at C-11, in addition to an acetoxy group at C-12, and the chemical shifts of C-11 and C-12 is 73.2 and 85.1 ppm, respectively. Isihippurol B (**26**) [27] is the same as compound (**24**) except for the presence of an acetoxy group at C-12. The chemical shift is more downfield from 83.6 to 86.2 ppm. 5β,6β-Epoxygorgostane-3β,7α,11α,12β-tetrol 11-acetate (**27**) [23] has a dihydroxy substitution at C-7 and C-12, in addition to an acetoxy group at C-11, with a chemical shift of 67.2, 82.6, and 77.5 ppm, respectively. 5β,6β-Epoxygorgostane-1α,3β,11α,12β-tetrol 11-acetate (**28**) [23] is the same as the previous compound except for the presence of a hydroxyl group at C-1 instead of C-7 with a chemical shift of 74.0 ppm. Moreover, the downfield chemical shift of C-2 is from 30.9 to 37.9 ppm, and the upfield chemical shift of C-3 is from 68.9 to 63.7 ppm. 5β,6β-Epoxygorgostane-3β,7α,11α,12β-tetrol 12-acetate (**29**) [22] has a dihydroxy substitution at C-7 and C-11, in addition to an acetoxy group at C-12, with a chemical shift of 67.4, 73.0, and 84.9 ppm, respectively. 5β,6β-Epoxyorgostane-1α,3β,11α,15α-tetrol 11,15-diacetate (**30**) [22] has a hydroxyl group at C-1 and di-acetoxy substitution at C-11 and C-15, with a chemical shift of 74.2, 72.4, and 75.2 ppm, respectively. 5β,6β-Epoxygorgostane-3β,7α,11α, 12β, 15α-pentol 12,15-diacetate (**31**) and 5β,6β-epoxygorgostane-3β,7α,11α, 12β,15α-pentol 11,15-diacetate (**32**) [22] have similar structures except for the substitution at C-11 and C-12. The first compound has hydroxyl and acetoxy groups at C-11 and C-12, with a chemical shift of 73.5 and 84.0 ppm, respectively. But the latter compound has acetoxy and hydroxyl groups at C-11 and C-12, with a chemical shift of 77.3 and 82.3 ppm, respectively.

### 4.3. 5,6-Dihydroxygorgostane

5,6-Dihydroxygorgostane is characterized by the presence of two hydroxyl groups at C-5 with an α orientation and C-6 with a β orientation (Figure 5 and Table 6). In ^13^C-NMR, C-5 and C-6 of xeniasterol C (**33**) showed characteristic signals at 75.9 and 76.3 ppm, respectively. The chemical shifts of C-5 and C-6 of 5, 6-dihydroxy gorgostane consequently changed according to the oxygenation pattern on A and B rings. The substitutions, mainly hydroxyl or acetoxy groups, usually occur at C-1, C-3, C-7, C-9, C-12, and C-20. Xeniasterol C (**33**) [28] is the parent compound of this group, which has only one additional hydroxyl group at C-3. Sarcoaldosterol A (**34**) [29] is the same as the previous compound except for an additional hydroxyl group at C-11, whose chemical shift is 68.5 ppm. Furthermore, the neighboring carbons, C-9 and C-12, are more downfield shifted at 53.1 and 53.0 ppm, respectively. 3β-Acetoxygorgostane-5α,6β, 11α-triol (**35**) [30] has an acetoxy group at C-3. The chemical shift of C-3 is downfield to 73.0 ppm due to the presence of an acetate group. Xeniasterol D (**36**) [28] has an acetoxy group at C-7 with a chemical shift of 76.2 ppm. Gorgostane-3β,5α,6β,11α-tetrol 11-acetate (**37**) [31] has an acetoxy group at C-11 with a chemical shift of 71.6 ppm. Gorgostane-3β,5α,6β, 11α, 20(S)-pentol 3-acetate (**38**) [32] is the same as compound (**35**), except the presence of a hydroxyl group at C-20 with an α orientation has more downfield shift (76.3 ppm). Gorgostane-3β,5α,6β,9α,11α-pentol (**39**) [10] is the same as compound (**34**), except the presence of a hydroxyl group at C-9 with an α orientation has more downfield shift (80.8 ppm); moreover, the neighboring carbons, C-8 and C-10, shift downfield at 33.2 and 43.5 ppm, respectively. 11α-Acetoxygorgostane-3β,5α,6β,12α-tetrol (**40**) [12] has an acetoxy group of α orientation at C-11 with a chemical shift of 73.4 ppm, in addition to an α-hydroxy group at C-12 with a chemical shift of 75.0 ppm. Compounds 12α-acetoxygorgostane-3β,5α,6β,11α-tetrol (**41**) [12] and gorgostane-3β,5α,6β,11α,12β-pentol 12-acetate (**42**) [22] have the same structure, except the configuration of an acetoxy group at C-12 has an α orientation at the first compound with a chemical shift of 80.5 ppm, but the latter has a β orientation with a more downfield chemical shift (85.8 ppm). The orientation of the acetate group at C-12 also affects the chemical shift value of C-18 methyl carbon, i.e., 12.7 ppm (**41**) and 9.9 ppm (**42**). Gorgostane-1α,3β,5α,6β,11α,12β-hexol 12-acetate (**43**) [22] has the same structure as the previous compound except for the addition of a hydroxyl group at C-1 with a chemical shift of 74.6 ppm, the downfield chemical shift of C-2 from 30.5 to 36.5 ppm, and the upfield chemical shift of C-3 from 66.8 to 63.0 ppm.

### 4.4. 9,11-Secogorgostane

9,11-Secogorgostane is characterized by the opening of ring C at C-9 and C-11. It is characterized by the presence of a ketone carbonyl carbon at C-9 and a primary alcohol group at C-11 (Figure 6 and Table 7). In ^13^C-NMR, C-9 and C-11 of 3β,11-dihydroxy-9,11-secogorgost-5-en-9-one (**44**) showed characteristic signals at 217.4 and 60.0 ppm, respectively. The chemical shifts of C-9 and C-11 of 9,11-secogorgostane consequently changed according to substitutions. Compounds 3β,11-dihydroxy-9,11-secogorgost-5-en-9-one (**44**), 3β,11,24-trihydroxy-9,11-secogorgost-5-en-9-one (**45**), and ameristerol A (**46**) [33,34,35] have the same structure and substitutions with a double bond between C-5 and C-6 except for the second compound containing a hydroxyl group at C-24 with a chemical shift of 74.5 ppm and the latter compound containing an additional double bond between C-24 and C-28 with a characteristic chemical shift of 161.4 and 105.3 ppm, respectively. Compounds 3β, 11-dihydroxy-5β, 6β-epoxy-9,11-secogorgostan-9-one (**47**), and 5α,6α-epoxy-3β,11-dihydroxy-9,11-secogorgostan-9-one (**48**) [36,37] are similar, except the first compound has a 5β, 6β-epoxy group at C-5 and C-6, and the others a 5α, 6α-epoxy group with a chemical shift of 65.4, 58.2 ppm and 60.9, 60.0 ppm, respectively. 3β,7β,11-trihydroxy-5α,6α -epoxy-9,11-secogorgostan-9-one (**49**) and 5α,6α-epoxy-1β,3β,11-trihydroxy-9,11-secogorgostan-9-one (**50**) [9,37] have the 5β, 6β-epoxy groups but with different substitutions in C-7 and C-1 with chemical shifts of 67.0 and 69.9 ppm, respectively, which affect the chemical shift of the neighboring carbons. Klyflaccisteroid F (**51**) [13] is a 9,11-secogorgost-5-ene skeleton containing a carboxyl group at C-11 with a chemical shift of 174.4 ppm. On the other hand, klyflaccisteroid K (**52**) [38] is a 5α,8α-epidioxy-9, 11-secogorgostane containing a double bond between C-6 and C-7 with a chemical shift of 141.2 and 130.2 ppm, respectively. Moreover, the chemical shift of C-3 is more upfield (65.9 ppm) than other 9, 11-secogorgostanes due to the presence of a 5α,8α-epidioxy group in this compound. Leptosterol C (**53**) [39] is a 9, 11-secogorgost-5-ene structure with a 23-demethyl side chain containing 29 carbons. Moreover, the chemical shift of C-24 is more upfield (44.8 ppm) than other 9, 11-secogorgostanes due to the absence of C-23 on the side chain cyclopropane moiety.

### 4.5. 23-Demethylgorgostane

23-Demethylgorgostane is characterized by the presence of only 29 carbon atoms and the lack of a methyl group that arises from the cyclopropane moiety at C-23 (Figure 7 and Table 8 and Table 9). In ^13^C-NMR, the chemical shift values for the cyclopropane moiety (C-22, C-23, and C-29) in 23-demethylgorgostane (25.2, 24.1, and 10.5 ppm) are different from those of gorgostane (C-22, C-23, and C-30) with a 24-methyl group (32.0, 26.1, and 21.3 ppm). This group could be 5, 6-epoxy, 5, 6-dihydroxy, or 5, 8-epidioxy gorgostane groups with different substitutions, mainly hydroxyl, carbonyl, and acetoxy or other groups such as chloride. These substitutions usually occur at C-1, C-2, C-3, C-4, C-7, and C-11. Stoloniferone M (**54**) [40] is a 23-demethylgorgostane containing a carbonyl group at C-1 and a hydroxyl group at C-3 with a 5, 6-β-epoxy gorgostane skeleton. The presence of a carbonyl group at C-1 leads to a high downfield ^13^C-NMR chemical shift in C-2 and C-10. Further, 5, 6-β-epoxy has an upfield chemical shift of up to 1 ppm more than 5, 6-α-epoxy gorgostane. Compounds 5α,6α-epoxy-23-demethylgorgost-8-ene-3β, 7α-diol (**55**) and 5α,6α-epoxy-23-demethylgorgost-8(14)-ene-3β, 7α-diol (**56**) [41] have the same substitutions except for the carbons of the double bond between C-8, C-9 and C-8, and C-14, respectively. The carbons of double bonds also downfield chemical shifts on neighboring carbons. 5α,8α-Epidioxygorgostane contains a double bond at C-6 and C-7 or may have an additional double bond at C-9 and C-11, such as 5α, 8α-epidioxy- 23-demethylgorgosta-6,9(11)-dien-3β-ol (**59**) [43]. In addition, compound 5α,8α-epidioxy-23,24-didemethylgorgost-6-ene-3β-ol (**58**) [42] has 28 carbon atoms due to 23, 24-didemethylgorgostane. Stoloniferone Q, D, J, S,yonarasterol C, I and F (**60-66**) [11,40,44,45,46] contain a carbonyl group at C-1 with a double bond between C-2 and C-3 except for stoloniferone S, which has a double bond between C-3 and C-4. Furthermore, stoloniferone Q has an additional bond between C-4 and C-5. The chemical shift of a carbonyl group at C-1 is variable due to the substitutions of neighboring carbons. C-1 at stoloniferone Q and S is a downfield shift to 212.5 and 212.0, while other compounds upfield shift from 208.7 to 204.4. Moreover, C-5 and C-6 (5, 6-epoxy group) in stoloniferone D are chemically shifted more upfield than 5, 6-dihydroxy groups or other substitutions in C-5 and C-6.

### 4.6. Miscellaneous Gorgostane

Miscellaneous Gorgostane has an undefined chemical structure (Figure 8 and Table 10). Compounds 5α,8α-epidioxygorgost-6-en-3β-ol (**67**) and 5α,8α-epidioxygorgosta-6,9(11)-dien-3β-ol (**68**) [47] have the same chemical structure except the later compound has an additional double bond between C-9 and C-11, which, due to a downfield chemical shift on neighboring carbons, cause both compounds to have a 5α,8α-epidioxy group and double bond between C-6 and C-7. Compounds 3α,5β-dihydroxygorgostan-6-one (**69**) and 1α,3β,5β,11α-tetrahydroxygorgostan-6-one (**70**) [48,49] are related to each other and have a 5β-hydroxy and carbonyl group at C-6, but the latter compound has α-dihydroxy substituted in C-1 and C-11. Furthermore, the orientation of the hydroxyl group at C-3 of the first compound is α, but the latter is β. These substitutions affect the chemical shift of both C-3 and C-6: 66.1, 212.3 ppm in the first compound and 68.0, 210.5 ppm in the latter. Compounds dissesterol (**71**) and gorgost-4-en-3-one (**75**) [50,51] have a double bond between C-4 and C-5, but compound (**71**) has 3α, 6β-dihydroxy substitution at C-3 and C-6, in contrast to the latter compound, which has a carbonyl group at C-3. Ameristerenol A and B (**72**, **73**) [35] are 9,11-secosterols and possess a seven-membered cyclic enol-ether in ring C with two double bonds in ring B between C-5, C-6 and C-8, and C-9, but compound (**73**) has an additional acetoxy group at C-3. Furthermore, the chemical shift of C-3 is more downfield to 71.9 and 74.0 ppm, respectively, similar to the gorgost-5-ene group. Compound klyflaccisteroid L (**74**) [38] has an unusual 11-norsteroid skeleton and is the first example of an 11-oxasteroid isolated from natural sources; it is a trihydroxy with substituted 3β, 7α, and 9α at C-3, C-7, and C-9, respectively. The chemical shift of hemiketal carbon C-9 is more downfield to 98.2 ppm.

## 5. Conclusions

This review provides an exploration of the structural diversity of gorgostane-type steroids isolated from marine sources and ^13^C-NMR spectroscopic data, which are considered an added value to the structural identification of gorgostane derivatives. Further investigation of this class of biological activity explaining its mechanisms of action in treating different diseases is required. These studies will assist in the discovery and development of new drugs from natural sources.

## Figures and Tables

**Figure 1 marinedrugs-20-00139-f001:**
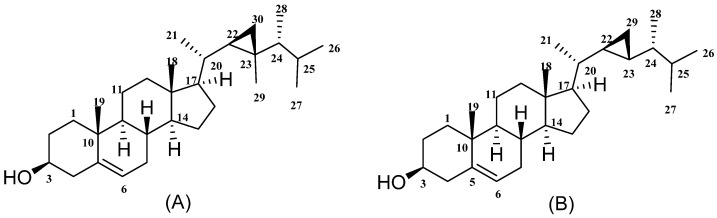
Gorgosterol (**A**) and 23-demethylgorgosterol (**B**).

**Figure 2 marinedrugs-20-00139-f002:**
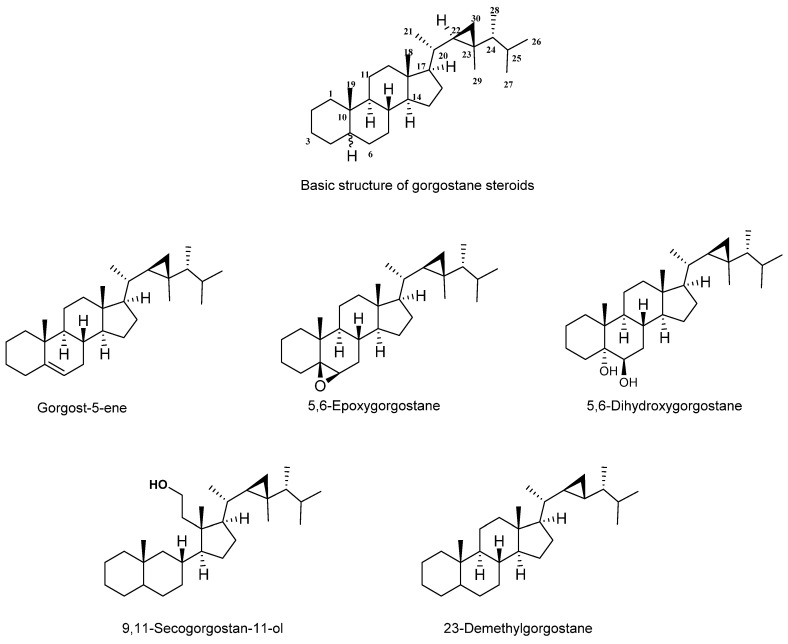
The basic skeleton of gorgostane steroids and their types.

**Figure 3 marinedrugs-20-00139-f003:**
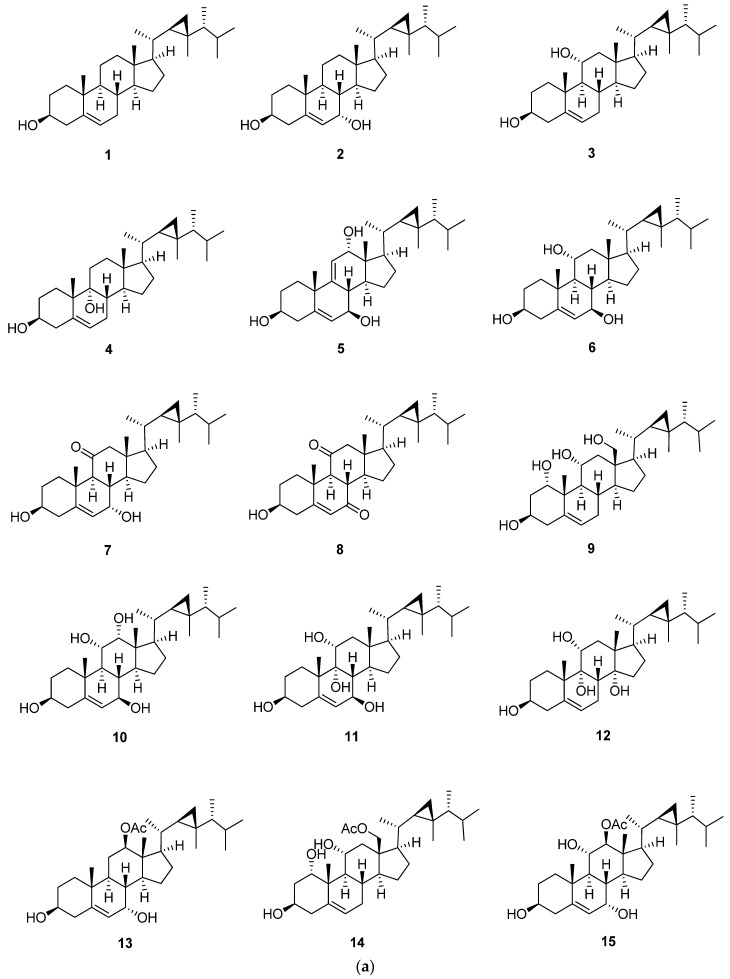
(**a**) Structures of isolated gorgost-5-ene steroids (**1**–**15**); (**b**) Structures of isolated gorgost-5-ene steroids (**16**–**22**).

**Figure 4 marinedrugs-20-00139-f004:**
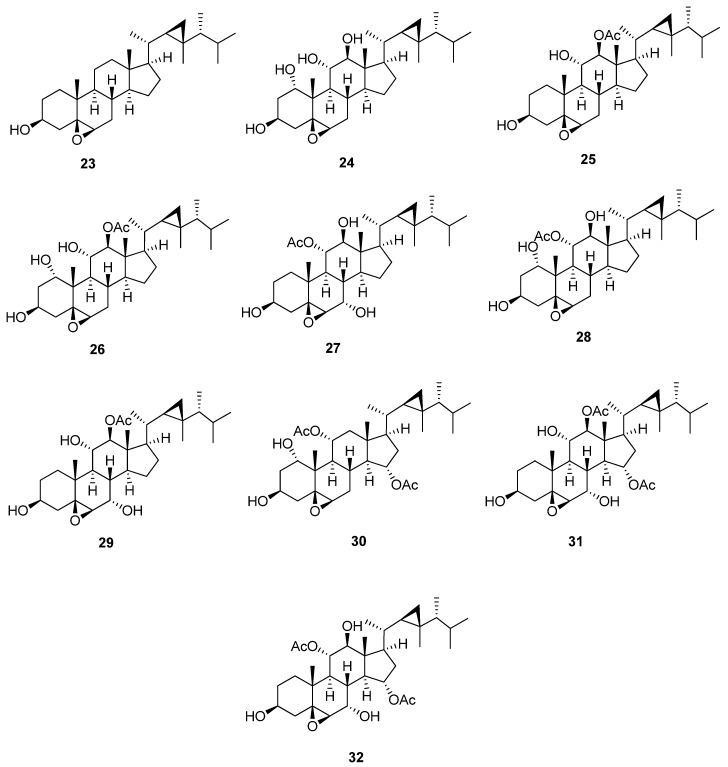
Structures of isolated 5,6-epoxygorgostane steroids (**23**–**32**).

**Figure 5 marinedrugs-20-00139-f005:**
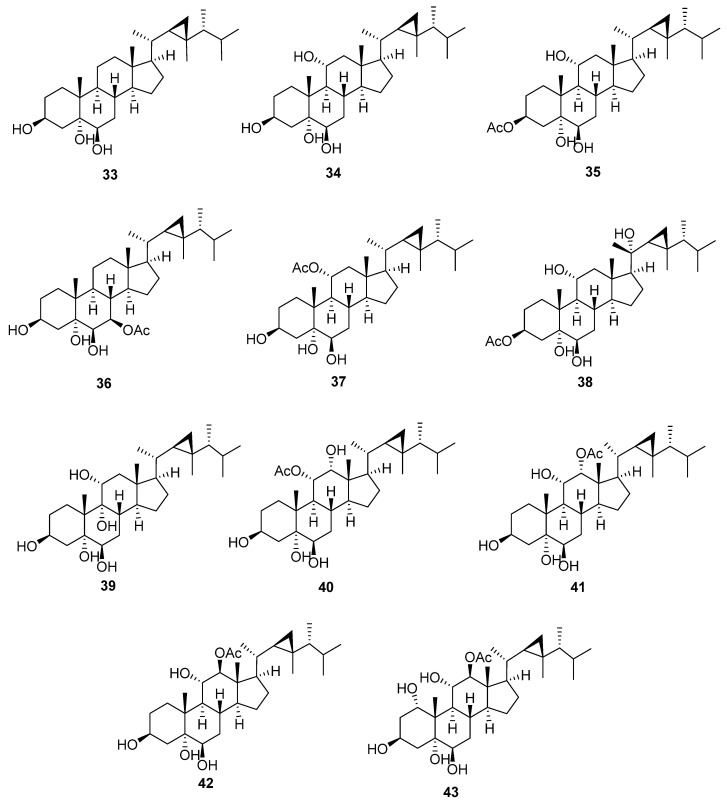
Structures of isolated 5,6-dihydroxygorgostane steroids (**33**–**43**).

**Figure 6 marinedrugs-20-00139-f006:**
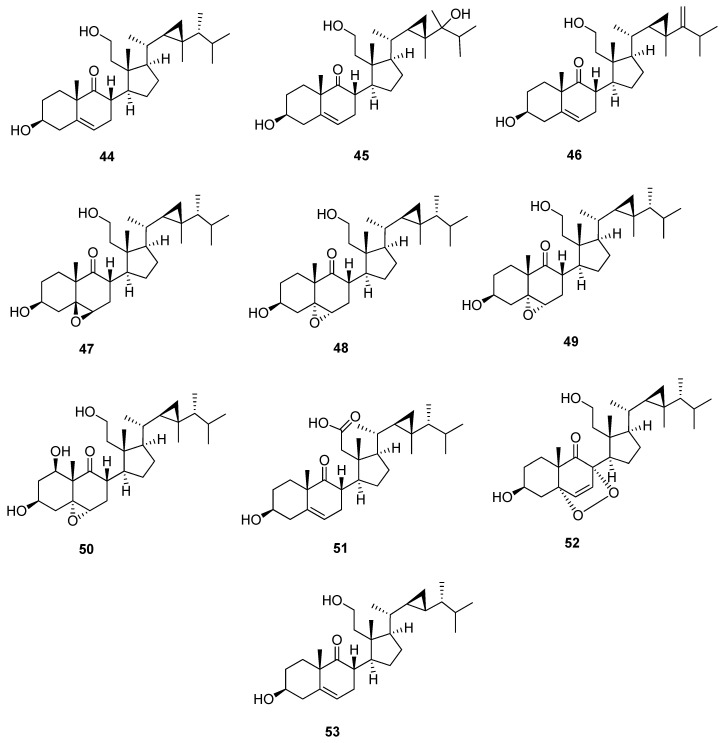
Structures of isolated 9,11-secogorgostane steroids (**44**–**53**).

**Figure 7 marinedrugs-20-00139-f007:**
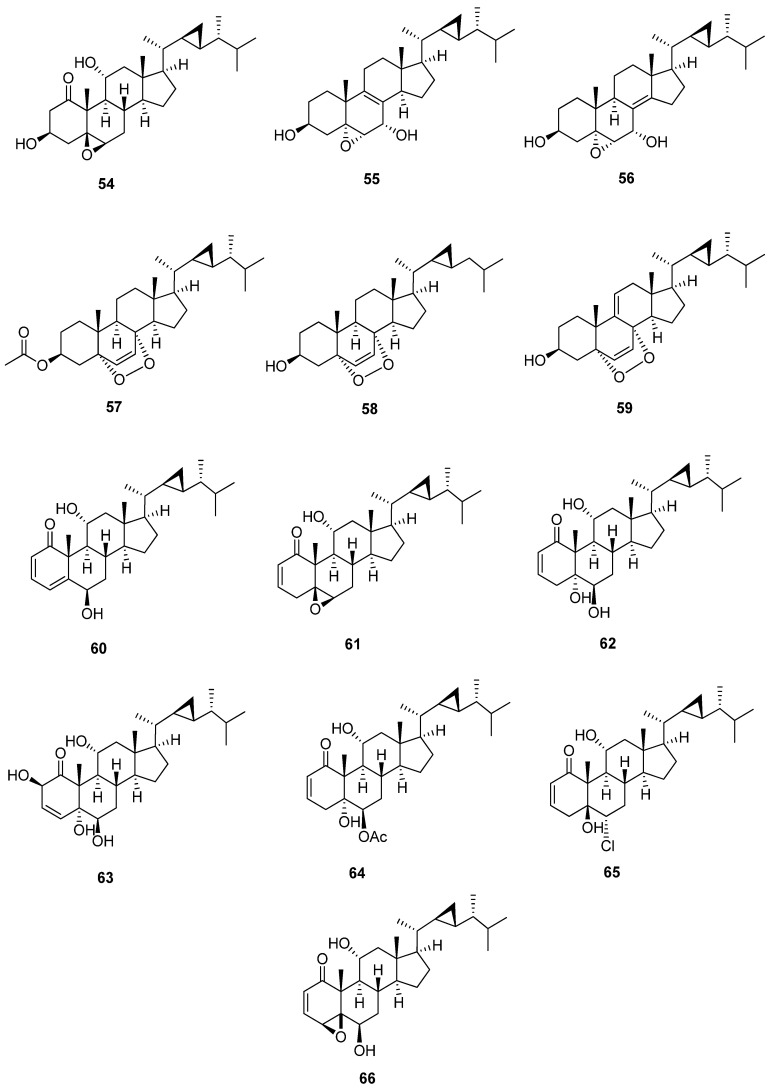
Structures of isolated 23-demethylgorgostane steroids (**54**–**66**).

**Figure 8 marinedrugs-20-00139-f008:**
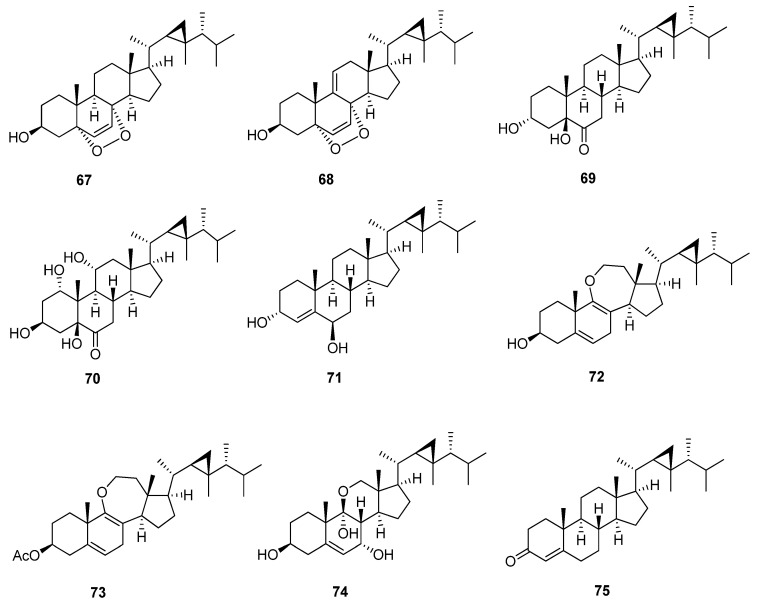
Structures of isolated miscellaneous gorgostane steroids (**67**–**75**).

**Table 1 marinedrugs-20-00139-t001:** Name and natural source of isolated gorgostane steroids (**1**–**75**).

No.	Name	Natural Source	References
**I. Gorgost-5-ene:**			
**1**	Gorgosterol	*Alcyonium molle*	[16]
**2**	Crassumsterol	*Lobophytum crassum*	[17]
**3**	Gorgost-5-ene-3β, 11α-diol	*Sarcophyton* *crassocaule*	[18]
**4**	9-Hydroxygorgosterol	*Plexaurella grisea*	[5]
**5**	Klyflaccisteroid E	*Klyxum flaccidum*	[13]
**6**	Klyflaccisteroid G	*Klyxum flaccidum*	[19]
**7**	Klyflaccisteroid C	*Klyxum flaccidum*	[13]
**8**	Klyflaccisteroid D	*Klyxum flaccidum*	[13]
**9**	Gorgost-5-ene-1α,3β,11α,18-tetrol	*Sinularia dissecta*	[20]
**10**	Klyflaccisteroid H	*Klyxum flaccidum*	[19]
**11**	Klyflaccisteroid I	*Klyxum flaccidum*	[19]
**12**	9,11α,14-Trihydroxygorgosterol	*Plexaurella grisea*	[5]
**13**	12β-Acetoxy-7α-hydroxygorgosterol	*Capnellala certiliensis*	[21]
**14**	18-Acetoxygorgost-5-ene-1α,3β,11α-triol	*Sinularia dissecta*	[20]
**15**	Gorgost-5-ene-3β,7α,11α,12β-tetrol 12-acetate	*Isis hippuris*	[22]
**16**	12β-Acetoxy-7α,19-dihydroxygorgosterol	*Capnellala certiliensis*	[21]
**17**	Gorgost-5-ene-3β,7α,11α,12β-tetrol 11-acetate	*Isis hippuris*	[23]
**18**	3β-Acetoxy-1α,11α-dihydroxygorgost-5-en-18-oic acid	*Sinularia dissecta*	[20]
**19**	Gorgost-5-ene-3β,7α,11α,12β,15α-pentol 12,15-diacetate	*Isis hippuris*	[22]
**20**	Gorgost-5-ene-3β,7α,11α,12β,15α-pentol 11,15-diacetate	*Isis hippuris*	[22]
**21**	Gorgost-5-ene-1α,3β,7α,11α,12β-pentol 12-acetate	*Isis minorbrachyblasta*	[24]
**22**	Gorgosta-5,25-dien-3β-ol	*Lobophytum lobophytum*	[25]
**II. 5,6-Epoxygorgostane:**			
**23**	5,6β-Epoxygorgosterol	*Sinularia leptoclados*	[26]
**24**	5β, 6β-Epoxygorgostane-1α, 3β,11α,12β-tetrol	*Isis hippuris*	[23]
**25**	5β,6β-Epoxygorgostane-3β,11α,12β -triol 12-acetate	*Isis hippuris*	[22]
**26**	Isihippurol B	*Isis hippuris*	[27]
**27**	5β,6β-Epoxygorgostane-3β,7α,11α,12β-tetrol 11-acetate	*Isis hippuris*	[23]
**28**	5β,6β-Epoxygorgostane-1α,3β,11α,12β-tetrol 11-acetate	*Isis hippuris*	[23]
**29**	5β,6β-Epoxygorgostane-3β,7α,11α,12β -tetrol 12-acetate	*Isis hippuris*	[22]
**30**	5β,6β-Epoxyorgostane-1α,3β,11α,15α-tetrol 11,15-diacetate	*Isis hippuris*	[22]
**31**	5β,6β-Epoxygorgostane-3β,7α,11α, 12β, 15α-pentol 12,15-diacetate	*Isis hippuris*	[22]
**32**	5β,6β-Epoxygorgostane-3β,7α,11α, 12β,15α-pentol 11,15-diacetate	*Isis hippuris*	[22]
**III. 5,6-Dihydroxygorgostane:**			
**33**	Xeniasterol C	*Xenia* sp.	[28]
**34**	Sarcoaldosterol A	*Sarcophyton* sp.	[29]
**35**	3β-Acetoxygorgostane-5α,6β, 11α-triol	*Heteroxenia fuscescens*	[30]
**36**	Xeniasterol D	*Xenia* sp.	[28]
**37**	Gorgostane-3β,5α,6β,11α-tetrol 11-acetate	*Heteroxenia ghardaqensis*	[31]
**38**	Gorgostane-3β,5α,6β, 11α, 20(S)-pentol 3-acetate	*Xenia umbellata*	[32]
**39**	Gorgostane-3β,5α,6β,9α,11α-pentol	*Eunicea laciniata*	[10]
**40**	11α-Acetoxygorgostane-3β,5α,6β,12α-tetraol	*Sarcophyton* sp.	[12]
**41**	12α-Acetoxygorgostane-3β,5α,6β,11α-tetraol	*Sarcophyton* sp.	[12]
**42**	Gorgostane-3β,5α,6β,11α,12β-pentol 12-acetate	*Isis hippuris*	[22]
**43**	Gorgostane-1α,3β,5α,6β,11α,12β-hexol 12-acetate	*Isis hippuris*	[22]
**IV. 9,11-Secogorgostane:**			
**44**	3β,11-Dihydroxy-9,11-secogorgost-5-en-9-one	*Parerythropodium fulvum*	[33]
**45**	3β,11,24-Trihydroxy-9,11-secogorgost-5-en-9-one	*Pseudopterogorgia* sp.	[34]
**46**	Ameristerol A	*Pseudopterogorgia americana*	[35]
**47**	5β,6β-Epoxy-3β,11-dihydroxy-9,11-secogorgostan-9-one	*Cespitularia* *hypotentaculata*	[36]
**48**	5α,6α-Epoxy-3β,11-dihydroxy-9,11-secogorgostan-9-one	*Pseudopterogorgia americana*	[37]
**49**	5α,6α-Epoxy-3β,7β,11-trihydroxy-9,11-secogorgostan-9-one	*Lobophytum* sp.	[9]
**50**	5α,6α-Epoxy-1β,3β,11-trihydroxy-9,11-secogorgostan-9-one	*Pseudopterogorgia americana*	[37]
**51**	Klyflaccisteroid F	*Klyxum flaccidum*	[13]
**52**	Klyflaccisteroid K	*Klyxum flaccidum*	[38]
**53**	Leptosterol C	*Sinularia leptoclados*	[39]
**V. 23-Demethylgorgostane:**			
**54**	Stoloniferone M	*Clavularia viridis*	[40]
**55**	5α,6α-Epoxy-23-demethylgorgost-8-ene-3β, 7α-diol	*Pinnigorgia* sp.	[41]
**56**	5α,6α-Epoxy-23-demethylgorgost-8(14)-ene-3β, 7α-diol	*Pinnigorgia* sp.	[41]
**57**	5α,8α-Epidioxy-23-demethylgorgost-6-ene-3β-yl acetate	*Sinularia maxima*	[42]
**58**	5α,8α-Epidioxy-23,24-didemethylgorgost-6-ene-3β-ol	*Sinularia maxima*	[42]
**59**	5α,8α-Epidioxy-23-demethylgorgosta- 6,9(11)-dien-3β-ol	*Sinularia gaweli*	[43]
**60**	Stoloniferone Q	*Clavularia viridis*	[40]
**61**	Stoloniferone D	*Clavularia viridis*	[44,45]
**62**	Stoloniferone J	*Clavularia viridis*	[40]
**63**	Stoloniferone S	*Clavularia viridis*	[11]
**64**	Yonarasterol C	*Clavularia viridis*	[44]
**65**	Yonarasterol I	*Clavularia viridis*	[46]
**66**	Yonarasterol F	*Clavularia viridis*	[44]
**VI. Miscellaneous gorgostane:**			
**67**	5α,8α-Epidioxygorgost-6-en-3β-ol	*Sinularia flexibilis*	[47]
**68**	5α,8α-Epidioxygorgosta-6,9(11)-dien-3β-ol	*Sinularia flexibilis*	[47]
**69**	3α,5β-Dihydroxygorgostan-6-one	*Sinularia* sp.	[48]
**70**	1α,3β,5β,11α-Tetrahydroxygorgostan-6-one	*Isis hippuris*	[49]
**71**	Dissesterol	*Sinularia dissecta*	[50]
**72**	Ameristerenol A	*Pseudopterogorgia americana*	[35]
**73**	Ameristerenol B	*Pseudopterogorgia americana*	[35]
**74**	Klyflaccisteroid L	*Klyxum flaccidum*	[38]
**75**	Gorgost-4-en-3-one	*Sinularia dissecta*	[51]

**Table 2 marinedrugs-20-00139-t002:** ^13^C-NMR data of isolated gorgost-5-ene steroids (**1**–**8**).

Carbon No.	1 ^a^	2 ^b^	3 ^c^	4 ^d^	5 ^b^	6 ^e^	7 ^e^	8 ^e^
1	37.3	37.0	31.1	29.6	34.6	39.0	35.9	35.4
2	31.7	31.3	31.5 ^f^	32.7	31.6	32.0	31.8	31.3
3	71.8	71.3	71.0	70.6	71.4	71.6	71.3	71.3
4	42.3	42.0	43.3	43.9	41.4	42.2	42.4	42.6
5	140.8	146.2	139.2	139.9	140.5	143.4	145.1	164.4
6	121.7	123.8	121.6	121.4	125.9	125.6	121.0	124
7	31.9	65.3	27.0	27.8	74.9	72.9	64.7	198.2
8	32.0	37.6	34.5 ^g^	35.2	44.6	40.2	40.5	46.2
9	50.2	42.3	49.5 ^h^	73.7	149.0	54.5	53.5	55.4
10	36.5	37.4	43.0	43.2	38.4	38.0	36.2	37.4
11	21.2	20.7	69.3	27.3	121.1	69.2	214.1	211.6
12	39.9	39.2	46.9	36.1	71.8	51.4	58.6	58.6
13	42.8	41.6	42.8	42.6	46.1	44.0	46.0	46.5
14	56.6	49.3	49.4	50.1	45.3	55.2	46.5	48.0
15	24.5	24.5	24.1	24.6	27.5	26.7	23.6	24.7
16	28.2	28.3	28.3	28.8	28.8	28.8	28.4	28.3
17	57.9	57.7	57.6 ^h^	58.1	48.1	57.0	57.9	58.0
18	11.9	11.6	12.1	11.4	11.2	12.9	14.1	13.9
19	19.4	18.2	22.1	23.0	27.1	18.7	26.7	25.8
20	35.3	35.4	35.2 ^g^	35.7	35.4	35.1	35.2	35.3
21	21.1	21.1	21.1	21.6	20.9	21.2	20.8	20.8
22	32.1	32.2	27.8	32.5	31.8	31.8	31.8	31.8
23	25.8	25.8	25.8	26.0	25.8	25.9	25.9	25.9
24	50.8	50.8	50.8 ^h^	50.9	50.7	50.8	50.7	50.7
25	32.1	32.0	32.0 ^f^	32.3	32.0	32.0	32.0	31.9
26	21.5	21.5	21.4	21.7	21.5	21.5	21.5	21.5
27	22.2	22.2	21.9	22.4	22.2	22.2	22.1	22.1
28	15.4	15.4	15.4	15.7	15.5	15.4	15.5	15.5
29	14.3	14.3	14.3 ^i^	14.4 ^j^	14.3	14.3	14.3	14.3
30	21.3	21.3	21.2 ^i^	21.5 ^j^	21.4	21.3	21.3	21.3

(^a^: 75 MHz in CDCl_3_; ^b^: 125 MHz in CDCl_3_; ^c^: 22.5 MHz in CDCl_3_; ^d^: 100 MHz in C_5_D_5_N; ^e^: 100 MHz in CDCl_3;_ and ^f,g,h,i,j^: These assignments are different in the original report and reassigned in this report by careful comparison).

**Table 3 marinedrugs-20-00139-t003:** ^13^C-NMR data of isolated gorgost-5-ene steroids (**9**–**16**).

Carbon No.	9 ^a^	10 ^b^	11 ^c^	12 ^d^	13 ^e^	14 ^a^	15 ^a^	16 ^e^
1	74.5	38.9	30.8	30.9	36.9	74.5	38.3	33.5
2	38.3	31.7	31.6	32.6	31.2	38.3	31.5	31.5
3	66.4	71.6	70.5	70.9	71.2	66.4	71.0	71.1
4	42.9	42.2	42.7	44.5	41.8	42.2	41.9	41.9
5	138.8	143.4	138.9	139.8	146	138.7	145.7	141.3
6	124.4	125.6	126.7	121.2	123.9	124.4	123.1	128.6
7	32.8	72.7	69.1	23.5	64.9	32.6	65.1	64.7
8	32.1	39.9	42.8	36.5	36.5	32.0	36.5	37.9
9	48.2	48.1	77.5	77.7	41.5	48.3	48.8	41.6
10	42.2	37.4	43.0	44.2	37.5	42.9	38.8	42.2
11	68.1	70.4	68.7	69.3	27.0	67.8	72.4	27.5
12	46.3	77.5	46.4	40.8	80.8	46.6	84.6	80.8
13	47.9	47.2	43.2	48.6	46.1	46.4	46.9	46.4
14	55.0	46.0	48.9	82.6	57.4	55.1	48.3	57.4
15	24.5	26.0	26.3	27.8	23.6	24.5	23.7	23.4
16	28.4	28.1	28.4	32.7	27.9	28.4	28.4	27.9
17	57.9	48.9	57.2	52.5	48.3	57.9	58.2	49.4
18	61.5	12.0	11.7	16.9	9.0	63.2	9.5	9.2
19	19.3	18.4	20.8	22.4	18.1	19.3	17.6	63.1
20	35.7	34.8	35.7	35.4	33.6	35.5	33.5	33.6
21	21.9	20.5	20.7	21.6	22.2	21.4	22.4	22.2
22	31.9	31.8	31.9	32.2	30.6	31.9	30.9	30.6
23	25.9	25.9	25.5	26.0	25.3	25.9	25.7	25.3
24	50.7	50.7	50.7	50.9	50.6	50.7	50.5	50.7
25	32.0	32.0	31.8	32.6	32.2	31.8	32.1	32.2
26	21.5	21.5	20.8	21.7	21.5	21.5	21.5	21.5
27	22.2	22.2	21.5	22.4	22.2	22.2	22.2	22.2
28	15.3	15.4	14.8	15.7	15.4	15.3	15.1	15.4
29	14.4	14.3	13.6	14.5 ^f^	13.8 ^g^	14.3	13.9	13.8 ^h^
30	21.3	21.3	20.7	21.4 ^f^	21.5 ^g^	21.3	21.3	21.5 ^h^
Ac1	---	---	---	---	170.7	171.0	173.3	170.8
	---	---	---	---	21.8	21.1	21.9	21.8

(^a^: 125 MHz in CDCl_3_; ^b^: 100MHz in CDCl_3_; ^c^: 100MHz in Acetone-d_6_; ^d^: 100 MHz in C_5_D_5_N; ^e^: 75.5 MHz in CDCl_3;_ and ^f,g,h^: These assignments are different in the original report and reassigned in this report by careful comparison).

**Table 4 marinedrugs-20-00139-t004:** ^13^C-NMR data of isolated gorgost-5-ene steroids (**17**–**22**).

Carbon No.	17 ^a^	18 ^b^	19 ^a^	20 ^a^	21 ^a^	22 ^c^
1	37.5	70.0	38.2	37.2	74.1	37.3
2	31.6	35.1	31.3	31.5	37.7	31.7
3	70.9	70.2	71.0	70.6	64.5	71.9
4	42.4	38.7	42.2	42.3	41.7	42.3
5	146.1	139.0	146.0	145.8	142.9	140.8
6	124	124.6	120.0	122.7	126.7	121.7
7	64.8	32.5	64.4	64.5	65.7	31.9
8	37.0	32.9	36.6	36.7	33.2	32.0
9	45.5	48.6	48.2	44.9	40.8	50.2
10	39.0	43.4	39.0	38.9	43.6	35.8
11	76.8	68.0	72.8	76.7	72.9	24.3
12	82.8	48.4	84.3	82.4	86.2	39.8
13	47.5	56.3	46.8	47.9	46.7	42.8
14	46.7	56.1	51.1	50.5	47.8	56.7
15	23.6	25.5	76.6	76.9	23.5	24.6
16	27.9	30.2	37.0	37.3	27.9	28.2
17	57.8	58.0	55.1	55.6	57.5	58.0
18	9.1	176.9	10.9	10.1	9.8	11.9
19	18.0	19.3	17.7	18.0	17.6	19.4
20	33.5	36.5	32.5	32.7	36.9	36.6
21	22.3	21.1	22.1	22.7	21.7	21.1
22	30.3	31.9	29.7	29.8	30.3	31.7
23	25.3	25.7	25.2	25.3	25.3	25.8
24	50.5	50.6	50.5	50.4	50.5	50.2
25	32.1	32.0	32.0	32.1	32.1	156.9
26	21.4	21.4	21.4	21.5	22.2	106.0
27	22.2	22.1	22.4	22.2	22.5	22.0
28	15.3	15.5	15.3	15.1	13.8	15.4
29	13.7	14.0	13.8	13.7	15.3	14.3
30	21.7	21.2	21.1	21.3	21.4	22.2
Ac1	172.8	170.0	170.1	170.0	173.4	---
	21.9	21.1	21.6	21.2	21.2	---
Ac2	---	---	172.5	173.1	---	---
	---	---	21.6	21.8	---	---

(^a^: 125 MHz in CDCl_3_; ^b^: 125 MHz in C_5_D_5_N; and ^c^: 150 MHz in CDCl_3_).

**Table 5 marinedrugs-20-00139-t005:** ^13^C-NMR data of isolated 5,6-epoxygorgostane steroids (**23**–**32**).

Carbon No.	23 ^a^	24 ^b^	25 ^a^	26 ^c^	27 ^a^	28 ^a^	29 ^a^	30 ^a^	31 ^a^	32 ^a^
1	37.2	73.8	38.7	74.5	37.7	74.0	38.6	74.2	38.0	37.6
2	31.1	37.6	31.1	40.0	30.9	37.9	31.2	37.9	30.9	30.8
3	69.5	63.4	69.3	64.3	68.9	63.7	69.0	63.8	69.1	68.8
4	42.3	42.1	42.6	44.6	42.2	42.2	42.2	42.2	42.0	42.0
5	62.9	63.3	63.1	65.1	63.7	62.2	64.3	61.9	63.7	63.4
6	63.7	64.5	63.0	64.6	64.0	63.6	64.4	63.2	63.5	63.4
7	32.6	32.2	31.6	32.8	67.2	31.8	67.4	31.2	66.9	66.9
8	29.9	28.0	28.0	29.0	32.8	27.7	33.1	27.1	32.5	32.1
9	51.4	45.6	56.9	47.3	45.0	44.5	48.4	44.9	47.9	44.9
10	34.9	40.8	35.9	42.3	35.1	40.7	35.4	40.6	35.3	35.2
11	22.1	73.8	73.2	72.9	77.5	77.4	73.0	72.4	73.5	77.3
12	39.9	83.6	85.1	86.2	82.6	82.9	84.9	45.5	84.0	82.3
13	42.8	47.3	47.0	48.2	47.7	48.0	46.9	43.4	46.8	48.0
14	56.1	53.4	53.7	54.5	46.9	52.7	48.6	58.3	52.0	51.3
15	24.4	23.8	23.6	24.8	22.9	23.6	23.1	75.2	76.2	76.5
16	28.2	28.1	27.7	28.5	27.7	27.7	28.3	38.4	37.1	37.4
17	58.0	58.3	57.6	58.1	58.0	58.0	58.2	55.3	55.4	55.9
18	11.8	9.1	10.0	10.9	9.1	9.0	9.7	13.4	11.1	10.2
19	17.0	15.8	15.6	16.7	16.3	16.0	16.0	16.0	15.7	16.3
20	35.2	33.5	33.2	34.1	33.3	33.3	33.2	34.6	32.0	32.6
21	21.1	22.5	22.3	22.5	22.4	22.4	22.6	20.3	22.3	22.3
22	32.1	30.6	30.3	31.1	30.2	30.2	30.7	31.8	29.8	29.9
23	25.8	25.4	25.3	25.9	25.3	25.2	25.7	25.8	25.3	25.3
24	50.8	50.7	50.6	51.4	50.5	50.5	50.6	50.6	50.5	50.5
25	32.0	32.0	32.1	33.0	32.1	32.1	32.2	31.7	32.5	32.5
26	21.3	21.5	21.4	22.6	21.4	21.4	21.5	21.3	21.2	21.9
27	22.2	22.3	22.2	23.0	21.9	22.2	22.3	22.1	22.2	22.3
28	15.4	15.3	15.4	16.5	15.3	15.3	15.2	15.2	15.2	15.0
29	14.3	13.8	13.8	13.8 ^d^	13.7	13.7	13.9	14.3	13.9	13.8
30	21.5	21.7	21.3	21.9	21.7	21.6	21.4	21.2	21.2	21.5
Ac1	---	---	173.2	171.8	172.7	172.3	173.7	169.5	169.8	169.8
	---	---	21.7	21.9	22.2	21.9	22.0	21.4	21.6	21.4
Ac2	---	---	---	---	---	---	---	170.6	172.9	172.9
	---	---	---	---	---	---	---	21.8	21.6	21.9

(^a^: 125 MHz in CDCl_3_; ^b^: 125 MHz in CDCl_3_:CD_3_OD, 5:1; ^c^: 125 MHz in C_5_D_5_N; and ^d^: These assignments are different in the original report and reassigned in this report by careful comparison).

**Table 6 marinedrugs-20-00139-t006:** ^13^C-NMR data of isolated 5,6-dihydroxygorgostane steroids (**33**–**43**).

Carbon No.	33 ^a^	34 ^b^	35 ^c^	36 ^a^	37 ^d^	38 ^e^	39 ^f^	40 ^d^	41 ^d^	42 ^g^	43 ^g^
1	32.5	35.6	35.1	32.2	34.2	26.9	30.4	33.7	33.9	33.7	74.6
2	33.3	35.4	28.2	33.4	35.1	34.0	31.4	31.3	31.2	30.5	36.5
3	67.4	67.4	73.0	67.1	68.9	70.8	67.9	67.1	67.5	66.8	63.0
4	42.8	43.5	38.5	42.5	40.1	37.5	41.8	41.3	41.3	40.2	40.3
5	75.9	76.8	77.2	76.6	76.5	76.8	78.8	76.6	76.7	76.2	77.7
6	76.3	76.5	76.5	77.5	76.0	76.2	76.8	76.0	76.0	75.1	73.9
7	35.7	35.8	35.2	76.2	34.6	34.5	29.7	34.4	34.0	33.6	34.1
8	31.1	30.1	30.3	36.0	32.0	29.0	33.2	29.4	28.9	28.6	28.8
9	46.0	53.1	52.8	45.3	51.9	52.7	80.8	42.7	46.3	50.5	45.8
10	39.2	41.1	41.0	38.5	40.0	39.9	43.5	39.6	39.6	39.8	41.9
11	21.8	68.5	69.0	22.0	71.6	68.6	70.5	73.4	70.7	72.6	71.2
12	40.8	53.0	53.0	40.6	52.5	51.9	47.3	75.0	80.5	85.8	86.5
13	43.6	44.0	44.4	44.4	44.7	43.6	44.2	46.7	46.0	47.0	46.9
14	56.5 ^h^	55.9	56.4	55.2	55.0	54.8	49.0	45.3	47.4	53.1	52.8
15	24.9	25.0	25.4	26.8	25.9	24.4	29.5	23.9	23.8	23.7	23.7
16	28.7	28.8	29.6	29.2	28.8	28.2	25.0	27.5	27.6	27.5	27.2
17	58.5 ^h^	58.0	59.4	57.5	58.0	57.8	59.1	49.2	50.1	57.4	57.0
18	12.5	13.4	13.5	12.4	13.1	13.0	12.8	11.8	12.7	9.9	9.9
19	17.2	17.8	17.4	17.8	16.8	16.9	20.0	16.7	16.7	16.3	15.6
20	35.6	35.6	36.5	35.5	35.3	76.3	36.4	35.1	34.6	33.8	32.7
21	22.4	22.4	21.6	22.4	22.3	21.1	21.5	20.1	20.6	22.0	21.8
22	32.2	32.3	33.4	32.2	32.0	31.9	32.9	31.9	31.9	30.2	29.7
23	26.0	25.9	26.8	26.0	26.1	25.8	26.7	25.9	25.9	25.1	24.8
24	50.9	50.9	52.2	50.9	51.8	50.7	52.2	50.8	50.8	50.1	50.4
25	32.5	32.3	33.3	32.6	32.2	32.0	33.3	32.0	32.0	31.9	31.9
26	21.5 ^i^	21.4	22.0	21.6 ^j^	21.7	22.2	21.9	21.5	21.5	21.5	21.4
27	15.6	21.7	22.1	15.6	21.5	21.5	22.6	22.2	21.3	22.0	22.0
28	21.6	15.7	16.0	21.6	15.7	15.5	15.9	15.4	15.5	15.1	15.1
29	14.4 ^i^	14.4	14.7	14.5 ^j^	14.4	14.2 ^k^	14.7	14.3	14.3	13.6	13.4
30	21.5	21.4	22.2	21.6	21.3	21.3 ^k^	22.1	21.3	21.3	21.0	20.8
Ac1	---	---	172.8	170.6	171.5	170.8	---	170.0	170.4	172.8	172.8
		---	21.4	21.8	22.8	21.4	---	22.1	22.2	21.5	21.8

(^a^: 22.5 MHz in C_5_D_5_N; ^b^: 100 MHz in C_5_D_5_N; ^c^: 150 MHz in CD_3_OD; ^d^: 100 MHz in CDCl_3_; ^e^:213 MHz in CDCl_3_; ^f^: 100 MHz in CD_3_OD; ^g^: 125 MHz in CDCl_3_:CD_3_OD,5:1; and ^h,i,j,k^: These assignments are different in the original report and reassigned in this report by careful comparison).

**Table 7 marinedrugs-20-00139-t007:** ^13^C-NMR data of isolated 9,11-secogorgostane steroids (**44**–**53**).

Carbon No.	44 ^a^	45 ^b^	46 ^c^	47 ^d^	48 ^e^	49 ^f^	50 ^e^	51 ^c^	52 ^c^	53 ^c^
1	31.0	31.1	31.0	28.6	29.9	31.2	69.9	31.0	28.3	31.1
2	30.7	30.5	30.8	30.5	34.8	29.9	35.0	30.6	29.3	30.8
3	72.0	71.0	71.5	68.0	69.5	69.2	69.4	71.3	65.9	71.4
4	40.6	40.4	40.6	38.6	39.9	40.0	40.3	40.5	35.3	40.6
5	140	140.4	140.1	65.4	60.9	62.7	61.0	140.3	83.8	140.4
6	122	121.2	121.5	58.2	60.0	65.2	60.4	121.1	141.2	121.5
7	32.2	32.5	32.8	26.2	32.0	67.0	31.8	32.8	130.2	33.0
8	43.2	43.0	43.5	38.8	41.6	49.3	41.5	44.1	86.6	43.5
9	217.4	217.7	217.6	214.8	214	213	214.1	218.5	205.9	217.6
10	36.0	48.3	48.3	46.6	46.7	45.4	46.4	48.4	44.7	48.4
11	60.0	58.8	59.1	58.8	59.0	59.1	59.1	174.4	59.4	59.4
12	40.4	40.3	40.5	41.3	39.6	40.7	39.4	43.5	43.7	40.2
13	45.0	45.6	45.4	45.9	45.5	45.8	45.6	45.6	45.9	45.6
14	41.7	41.6	41.6	45.3	40.2	43.0	40.4	42.9	45.7	41.6
15	24.4	24.1	24.3	22.6	26.0	23.0	25.8	24.9	24.9	24.4
16	27.5	27.8	26.8	28.2	27.9	28.1	27.7	27.1	29.3	26.7
17	50.3	50.2	50.7	50.7	50.0	50.3	50.2	51.1	53.1	50.2
18	17.0	17.3	17.3	18.1	17.0	17.8	17.2	16.8	20.3	17.2
19	22.9 ^g^	22.9	22.9	19.7	17.4	18.1	17.5	22.9	21.7	22.9
20	34.5	34.6	33.8	34.7	34.9	35.2	35.0	34.7	33.5	38.4
21	20.6	20.4	20.5	21.0	20.8	20.5	20.6	21.0	21.3	18.8
22	31.8	26.0	31.4	32.1	31.9	31.9	32.0	31.6	31.9	23.9 ^h^
23	25.8	28.6	26.5	25.9	25.9	25.9	25.8	25.9	25.9	24.4 ^h^
24	50.6	74.5	161.4	50.8	50.5	50.7	50.7	50.6	50.6	44.8
25	32.0	34.7	29.4	32.0	31.4	31.2	31.5	32.0	32.0	32.8
26	21.4 ^g^	17.2	24.2	22.3	22.3	22.0	22.2	21.4	21.4	18.5
27	15.3	17.0	24.2	21.4	21.5	21.5	21.4	22.2	22.2	20.7
28	21.6	23.9	105.3	15.2	15.2	15.3	15.3	15.3	15.1	15.7
29	14.2 ^g^	16.9	20.4	14.4	14.2	14.3	14.3	14.2	14.2	10.5
30	21.2	16.1	17.8	21.4	21.2	21.3	21.3	21.3	21.4	---

(^a^: 90 MHz in CDCl_3_; ^b^: 125 MHz in CDCl_3_+3 drops of CD_3_OD; ^c^: 100 MHz in CDCl_3_; ^d^: 75 MHz in CDCl_3_; ^e^: 125 MHz in CDCl_3_; ^f^: 150 MHz in CDCl_3;_ and ^g,h^: These assignments are different in the original report and reassigned in this report by careful comparison).

**Table 8 marinedrugs-20-00139-t008:** ^13^C-NMR data of isolated 23-demethylgorgostane steroids (**54**–**60**).

Carbon No.	54 ^a^	55 ^b^	56 ^b^	57 ^c^	58 ^c,d^	59 ^b^	60 ^a^
1	215.2	30.2	32.2	39.4	39.4	32.6	212.5
2	47.1	30.9	31.1	30.0	30.1	30.6	126.9
3	64.2	68.6	68.7	69.5	66.4	66.3	140.8
4	40.9	39.2	39.6	51.1	51.1	36.1	119.0
5	61.3	65.6	67.8	79.4	79.5	82.7	157.7
6	61.0	62.6	61.3	130.9	130.8	130.8	73.7
7	31.6	67.1	65.1	135.0	135.4	135.4	40.4
8	28.5	126.9	125.1	81.7	82.1	78.4	29.5
9	49.8	134.6	38.7	34.3	34.7	142.5	58.2
10	51.7	38.0	35.8	36.9	37.0	37.9	55.4
11	67.6	23.7	19.0	20.8	20.9	119.8	66.9
12	49.2	35.7	36.6	39.5	39.5	41.2	49.5
13	43.2	42.5	43.3	44.9	44.9	44.1	43.2
14	55.2	49.3	152.6	51.3	51.4	47.8	54.6
15	24.4	23.9	25.3	28.7	28.5	21.2	24.9
16	29.8	29.1	27.3	23.4	23.4	28.4	28.6
17	57.4	55.1	58.2	57.3	57.3	57.4	57.5
18	12.6	10.9	17.7	12.5	12.5	12.6	12.9
19	13.6	22.8	16.5	18.0	18.5	25.5	19.6
20	40.0	40.5	39.2	39.9	39.9	39.7	40.1
21	19.1	19.2	19.2	19.0	19.1	19.0	19.2
22	24.1 ^e^	24.0	24.0	24.1 ^e^	24.1 ^e^	24.2	24.1 ^e^
23	25.2 ^e^	25.2	25.1	25.1 ^e^	25.1 ^e^	25.1	25.2 ^e^
24	45.0	45.0	44.3	45.0	45.0	44.9	45.0
25	32.9	32.8	32.3	33.2	32.8	32.8	32.9
26	18.6	20.7	20.7	18.5	18.1	18.5	18.6
27	20.7	18.6	18.3	20.6	20.6	20.7	20.8
28	15.8	15.8	15.7	16.1	10.5	15.8	15.9
29	10.5	10.4	10.6	10.5	---	10.5	10.5
30	---	---	---	---	---	---	---
Ac1	---	---	---	170.0	---	---	---
	---	---	---	21.2	---	---	---

(^a^: 75 MHz in CDCl_3_; ^b^: 100 MHz in CDCl_3_; ^c^: 22.5 MHz in CDCl_3_; ^d^: These NMR data do not seem to support the proposed structure [55]; and ^e^: These assignments are different in the original report and reassigned in this report by careful comparison).

**Table 9 marinedrugs-20-00139-t009:** ^13^C-NMR data of isolated 23-demethylgorgostane steroids (**61**–**66**).

Carbon No.	61 ^a^	62 ^b^	63 ^c^	64 ^c^	65 ^c^	66 ^c^
1	207.9	208.7	212	207.7	205.3	204.4
2	128.7	128.9	78.5	129.1	128.6	131.8
3	147.3	142.1	126.4	141.0	140.1	138.7
4	34.0	36.3	141.7	36.4	31.7	54.1
5	62.3	77.9	83.9	76.9	78.2	65.1
6	63.7	74.7	66.9	75.3	66.1	73.3
7	30.5	32.9	34.9	29.3	38.8	37.3
8	28.7	28.4	27.7	29.2	33.9	28.7
9	50.6	47.2	50.0	47.1	54.3	57.2
10	50.1	54.3	49.2	53.7	54.5	49.6
11	67.1	68.4	67.0	68.5	66.8	66.4
12	50.8	51.2	48.4	51.1	48.8	48.8
13	43.4	43.6	43.0	43.5	43.5	43.4
14	55.4	55.6	54.0	55.6	54.4	54.3
15	24.3	24.0	24.6	24.1	28.3	24.4
16	28.7	28.5	28.4	28.6	24.1	28.4
17	57.5	57.5	57.4	57.3	57.2	57.3
18	12.9	13.1	13.0	13.5	12.8	12.7
19	14.9	15.1	20.1	14.6	9.3	13.5
20	40.0	39.8	40.1	40.1	39.9	40.0
21	19.0	19.1	19.2	19.0	19.1	19.1
22	24.0	24.0 ^d^	24.0 ^d^	23.9 ^d^	24.1 ^d^	24.1 ^d^
23	25.1	25.2 ^d^	25.2 ^d^	25.1 ^d^	25.1 ^d^	25.2 ^d^
24	45.0	45.0	44.9	44.9	44.9	44.9
25	32.9	33.0	32.8	32.8	32.8	32.8
26	18.6	18.6	18.5	18.6	18.5	18.5
27	20.7	20.7	20.7	20.7	20.7	20.7
28	15.9	15.9	15.8	15.8	15.8	15.8
29	10.5	10.5	10.5	10.4	10.5	10.5
30	---	---	---	-----	---	---
Ac1	---	---	---	21.3	---	---
	---	---	---	169.9	---	---

(^a^: 22.5 MHz in CDCl_3_; ^b^: 75 MHz in CDCl_3_; ^c^: 125 MHz in CDCl_3;_ and ^d^: These assignments are different in the original report and reassigned in this report by careful comparison).

**Table 10 marinedrugs-20-00139-t010:** ^13^C-NMR data of isolated miscellaneous gorgostane steroids (**67**–**75**).

Carbon No.	67 ^a^	68 ^a^	69 ^a^	70 ^a^	71 ^b^	72 ^c^	73 ^c^	74 ^c^	75 ^a^
1	34.7	34.7	33.6	70.4	36.5	34.0	33.9	27.0	36.1
2	30.1	30.1	27.5	37.4	28.5	31.2	27.3	30.9	34.4
3	66.5	65.4	66.1	68.0	67.3	71.9	74.0	70.7	200.0
4	37.0	36.1	37.2	37.3	128.1	41.0	37.0	41.4	124.2
5	82.2	82.7	84.2	84.0	147.1	138.2	137.1	140.4	172.1
6	130.8	130.8	212.3	210.5	73.6	118.6	119.6	125.1	32.4
7	135.4	135.4	41.5	41.1	38.9	30.3	30.3	71.2	33.2
8	79.5	78.5	35.8	36.0	30.1	115.0	115.0	43.8	36.0
9	51.1	142.5	44.9	50.2	54.1	156.6	156.4	98.2	54.2
10	37.0	37.0	40.0	50.6	36.7	39.3	39.4	43.0	38.9
11	29.8	119.8	21.5	66.3	20.8	68.0	68.0	---	21.4
12	39.5	41.4	39.0	49.0	39.7	46.4	46.4	73.8	40.1
13	45.2	44.2	42.1	43.3	42.8	42.8	42.9	43.7	43.2
14	51.5	51.6	54.0	55.4	55.8	50.4	50.7	46.1	58.2
15	23.5	23.4	23.8	24.5	24.2	24.4	24.7	26.1	24.8
16	28.3	28.3	27.5	28.0	28.0	27.9	27.9	28.3	28.6
17	58.2	57.8	56.8	57.8	57.8	58.4	58.5	54.2	56.1
18	12.6	12.7	12.2	12.7	11.8	12.2	12.2	10.2	12.4
19	18.2	25.5	19.9	14.6	21.1	21.5	21.5	20.9	17.7
20	34.7	34.9	34.9	35.2	35.1	35.4	35.4	34.7	35.6
21	21.5	21.1	21.2	21.0	21.1	21.7	21.7	21.0	21.5
22	31.9	31.8	31.0	31.9	31.8	32.0	32.3	31.7	32.4
23	26.0	25.9	25.0	25.9	25.6	25.9	26.2	25.9	26.2
24	50.7	50.7	50.0	50.7	50.6	50.7	50.8	50.7	51.8
25	32.0	32.0	31.2	32.0	32.0	32.1	32.2	32.0	32.4
26	21.5	21.5	22.2	21.5	21.2	21.5	21.5	21.5	21.9
27	22.2	22.2	21.4	22.1	21.8	22.2	22 2	22.1	22.5
28	15.4	15.5	15.4	15.5	15.2	15.3	15.4	15.5	15.8
29	14.3	14.3	14.0 ^d^	14.3	14.0	14.2	14.2	14.3	14.7
30	21.3	21.3	20.5 ^d^	21.3	20.9	21.4	21.4	21.3	21.7
Ac1	---	---	---	---	---	---	170.2	---	---
	---	---	---	---	---	---	21.4	---	---

(^a^: 125 MHz in CDCl_3_; ^b^: 125 MHz in CDCl_3_+CD_3_OD; ^c^: 100 MHz in CDCl_3;_ and ^d^: These assignments are different in the original report and reassigned in this report by careful comparison).

## Data Availability

Not applicable.

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
