# Peer review of "Chemical Review of Gorgostane-Type Steroids Isolated from Marine Organisms and Their 13C-NMR Spectroscopic Data Characteristics"

_marinedrugs, 2022, doi:10.3390/md20020139_

Round 1

Reviewer 1 Report

The review manuscript entitled “Chemical review of Gorgostane type steroids isolated from marine organisms and their 13C-NMR spectroscopic data characteristics” written by Abdelkarem and co-workers describes on all 13C-NMR chemical shift data of known the family of steroids to date. Treating gorgostane steroids seems suitable to a review article in marine drugs with a lot of information on detailed data of 13C-NMR chemical sifts for overall 75 compounds. In contrary, it is so difficult to understand or follow them. So, the reviewer suggests major modification of this review before publication. Please revise along the following points.

  1. Numbering of carbons are not required in Figure 2, but Figure 1. Present Figure 2 seems busy.
  2. Way of presenting the absolute configurations should be uniformed for all compounds, especially for C20, 22,23, in all figures. Or present S or R configurations for C22 and C23. They are quite confusing.
  3. It is so difficult to check structures and chemical shifts each other in text, figures, and tables. The reviewer recommends dividing Figure 3 along Table 1’s classification. The following tables should also be reproduced along it. And put them in suitable places to understand easier for readers.
  4. Descriptions on chemical sifts in text are quite bored. Down-field shift for C7 in compounds 5-7 against mother compound 1, or up-filed shift of C8 in compound 8 for unsaturated ketone against C11 ketone in the same compound, are ovious.
  5. Compound numbers in Table 1 and title in Table 2-10 should be presented in bold.
  6. Table 2-10; left column; carbon number or position.
  7. The authors should show preference in chemical sift variation for entire data and some discussion for reasonable interpretation. The reviewer wants to know what do the authors wants to claim.

Author Response

Dear Reviewer 1,

Manuscript ID: marinedrugs-1578220

Type of manuscript: Review

Title: Chemical review of Gorgostane type steroids isolated from marine organisms and their 13C-NMR spectroscopic data characteristics

Thank you very much for your fruitful comments and recommendations, the requested comments are already included and highlighted in the main manuscript, here, the responds to each reviewer comments separately.

 Reviewer 1:

Comments and Suggestions for Authors:

The review manuscript entitled “Chemical review of Gorgostane type steroids isolated from marine organisms and their 13C-NMR spectroscopic data characteristics” written by Abdelkarem and co-workers describes on all 13C-NMR chemical shift data of known the family of steroids to date. Treating gorgostane steroids seems suitable to a review article in marine drugs with a lot of information on detailed data of 13C-NMR chemical sifts for overall 75 compounds. In contrary, it is so difficult to understand or follow them. So, the reviewer suggests major modification of this review before publication. Please revise along the following points.

1- Numbering of carbons are not required in Figure 2, but Figure 1. Present Figure 2 seems busy.

Response:

Thank you very much for your comment, we modified Figure 1 and Figure 2 according to the reviewer’s suggestion.

2- Way of presenting the absolute configurations should be uniformed for all compounds, especially for C20, 22,23, in all figures. Or present S or R configurations for C22 and C23. They are quite confusing.

Response:

Thank you very much for your comment, we uniformed the configuration of gorgostane steroids especially for C-20, C-22, and C-23 based on the biogenetic point of view, as well as X-ray crystallography, the absolute configuration of the side chain is (20S, 22R and 23R) (Tanaka et al., 2002).

3- It is so difficult to check structures and chemical shifts each other in text, figures, and tables. The reviewer recommends dividing Figure 3 along Table 1’s classification. The following tables should also be reproduced along it. And put them in suitable places to understand easier for readers.

Response:

Thank you very much for your comment, gorgostane compounds are classified into six sub-classes, the number of compounds in each class is not uniform (22, 10, 11, 10, 13, and 9), if we divided these compounds according to sub-classes, the format of the figures will not be uniformed. According to the reviewer’s suggestion, we divided Figure 3 into five Figures, each Figure has fifteen compounds.

4- Descriptions on chemical sifts in text are quite bored. Down-field shift for C7 in compounds 5-7 against mother compound 1, or up-filed shift of C8 in compound 8 for unsaturated ketone against C11 ketone in the same compound, are ovious.

Response:

Thank you very much for your comment, we delete this paragraph as the reviewer’s suggested.

5- Compound numbers in Table 1 and title in Table 2-10 should be presented in bold.

Response:

Thank you very much for your comment, we modified compound numbers in Table 1 and the title in Table 2-10 as reviewer’s suggestion.

6- Table 2-10; left column; carbon number or position.

Response:

Thank you very much for your comment, we modified Table 2-10; left column; carbon number as reviewer’s suggestion.

7- The authors should show preference in chemical sift variation for entire data and some discussion for reasonable interpretation. The reviewer wants to know what do the authors wants to claim.

Response:

Thank you very much for your comment, in this review, we briefly discussed some of the characteristic 13C-NMR chemical shifts of six gorgostane sub-classes, which are considered an added value to structure identification of the gorgostane derivatives, the more details on identification and structure elucidation of these compounds have already in the original report with its reference.

Reviewer 2 Report

The authors have done a lot of work to bring together in this review article all the steroids that make up the gorgostane family. Divided into 5 subgroups, the structures were reported and the 13C-NMR data briefly discussed in order to bring out the characteristics of these 75 gorgostanes isolated from marine organisms (73 from corals and 2 from algals). For this reason, the manuscript fits very well in Marine Drugs.

- The chemical structures have been reported in an impeccable manner (without errors).

- Tables are numerous but necessary.

- Section 2 is very short and should be a little more elaborate if possible.

- At the end of the first paragraph of section 4, authors should use a short text to link to Figure 3, Table 1 and Tables 1-10 (there is no link currently). Briefly, Table 1 (names and natural sources of all gorgostanes), Figure 3 (chemical structures) and Tables 1-11 (13C-NMR data).

- Titles from references 4, 7, 16, 22 and 46 should not be capitalyzed.

- The manuscript is well written, but small errors will have to be corrected. Please see the file, marinedrugs-1578220-peer-review-v1 (reviewer corrections), showing the corrections on the text.

Author Response

Dear Reviewer 2,

Manuscript ID: marinedrugs-1578220

Type of manuscript: Review

Title: Chemical review of Gorgostane type steroids isolated from marine organisms and their 13C-NMR spectroscopic data characteristics

Thank you very much for your fruitful comments and recommendations, the requested comments are already included and highlighted in the main manuscript, here, the responds to each reviewer comments separately.

 Reviewer 2:

Comments and Suggestions for Authors:

The authors have done a lot of work to bring together in this review article all the steroids that make up the gorgostane family. Divided into 5 subgroups, the structures were reported and the 13C-NMR data briefly discussed in order to bring out the characteristics of these 75 gorgostanes isolated from marine organisms (73 from corals and 2 from algals). For this reason, the manuscript fits very well in Marine Drugs.

- The chemical structures have been reported in an impeccable manner (without errors).

- Tables are numerous but necessary.

- Section 2 is very short and should be a little more elaborate if possible.

Response:

Thank you very much for your comment, we improved this section (distribution of gorgosteroids among marine organisms) as the reviewer’s suggestion.

 - At the end of the first paragraph of section 4, authors should use a short text to link to Figure 3, Table 1, and Tables 1-10 (there is no link currently). Briefly, Table 1 (names and natural sources of all gorgostanes), Figure 3 (chemical structures) and Tables 1-11 (13C-NMR data).

Response:

Thank you very much for your suggestion, we use a short text to link to (Figures 3-7) and (Tables 2-11) in section 2 and 4 as reviewer’s suggestion, also, we shortened (Table 1, Figure 3, and Tables 1-11) as reviewer’s suggestion.

- Titles from references 4, 7, 16, 22 and 46 should not be capitalyzed.

Response:

Thank you very much for your comment, we corrected these typographical errors.

- The manuscript is well written, but small errors will have to be corrected. Please see the file, marinedrugs-1578220-peer-review-v1 (reviewer corrections), showing the corrections on the text.

Response:

Thank you very much for your fruitful comments and recommendations, we already revised this version of manuscript (marinedrugs-1578220-peer-review-v1) as editor and reviewer’s suggestion.

Author Response

Dear Reviewer 3,

Manuscript ID: marinedrugs-1578220

Type of manuscript: Review

Title: Chemical review of Gorgostane type steroids isolated from marine organisms and their 13C-NMR spectroscopic data characteristics

Thank you very much for your fruitful comments and recommendations, the requested comments are already included and highlighted in the main manuscript, here, the responds to each reviewer comments separately.

 Reviewer 3:

In this review, there are many useful information on 13C NMR data and structures on gorgostane steroids with unique side chain portion. Most of structural diversities are derived from structures on core steroid A-D ring system with several functional groups mainly by oxidation. In the text, typical chemical shift values for oxygenated carbons are mentioned, but there are no logical explanation on substitution effects. Please add the reference as below.

Carbon-13 Nuclear Magnetic Resonance Spectra of Hydroxy Steroids,

  1. Eggert et al, J. Org. Chem. 1976, 41, 71-78.

And some review on steroids and NMR data should be referred.

For example,

C-13 NMR-Studies. 69. C-13 NMR-Spectra of Steroids- Survey and Commentary,

Blunt, J. W. and Stothers, J. B., Org. Magn. Reson. 1977, 9, 439-464.

Steroids and NMR, Jaeger, M. and Aspers, R. L. E. G.. Ann. Rep. NMR Spect. 2022, 77, 115-258.

Response:

Thank you very much for your comment, these three references were already cited in the introduction of the manuscript, as reviewer’s suggestion.

In this review, 13C NMR data of 75 gorgostane steroids are summarized in Table 2.

Unfortunately, there are many misassignments on the Table 2. Please take care for evaluation of the original assignments. 13C NMR data are suitable for evaluation by comparison of analogous compounds.

Some of the reason would be derived from mistake of assignments in the original reports, but in the process of reviewing the 13C NMR data with careful comparison between analogous compounds, it would be revealed.

Please check listed below with carefully, and if there are mistaken in the original reports, please mention in the text or as comments.

C-29 and C-30 assignments for 1 is 14.3 and 21.3 ppm. It should be correct. But there are many reversed assignments for compounds 3, 4, 12,13, 16, 38, and 69.

Response:

Thank you very much for your comment, we checked C-29 and C-30 assignments for compounds 3, 4, 12,13, 16, 38, and 69, there is no contrast in our manuscript with those in the original reports, it may be interchangeable carbons between C-29 and C-30 assignments, but authors in the reported data not illustrated it, however, compounds 33 and 36 its carbons C-29 and C-30 authors in this report (Kitagawa et al., 1986) mentioned these carbons as interchangeable.

Some chemical shift values for C-29 assignments should be revised.

Compound 26: C-29 is 23.3 ppm.

Response:

Thank you very much for your comment, we checked C-29 assignment of compound 26, it is the same as the original report.

Compound 33: C-29 is 21.5ppm. (C-26 at 14.4 ppm should be this 21.5 ppm).

 Response:

Thank you very much for your comment, we checked compound 33, C-26 and C-29 are interchangeable carbons in the original report, and we corrected it in the manuscript and labeled the two carbons in Table 6 with (*).

Compound 36: C-29 is 21.6ppm. (C-26 at 14.5 ppm should be this 21.6 ppm).

Response:

Thank you very much for your comment, we checked compound 36, C-26 and C-29 are interchangeable carbons in the original report, and we corrected it in the manuscript and labeled the two carbons in Table 6 with (*).

Compound 44: C-29 is 22.9 ppm. (C-26 at 14.2 ppm should be assigned to C-29, but 22.9 ppm should be assigned to C-19, and 21.4 ppm of C-19 would be C-29). Based on comparison to the data for compounds 45 and 46.

Response:

Thank you very much for your comment, we checked C-29 assignment of compound 44, it is the same as the original report, but we found that, the difference in chemical shift due to the presence of hydroxyl group at C-24 in compound 45 and presence of double bond C-24 and C-28.

Assignments for compound 3 is confusing. Some assignments should be revised.

C-3 at 32.0 ppm and C-25 at 31.5 ppm should be exchanged.

C-9 at 57.6 ppm should be assigned to C-17, C-17 at 50.8 ppm should be assigned to C-24, And C-24 at 49.5 ppm should be assigned to C-9.

Response:

Thank you very much for your comment, we checked the assignment of compound 3, it is the same as the original report. Also, compound 3, its C-3 is 71.0 ppm, not 32.0 in the manuscript as in the original report.

Compound 33: C-17 at 56.5 ppm should be exchanged to C-14 58.5 ppm.

Response:

Thank you very much for your comment, we checked C-17 and C-14 assignment of compound 33, it is the same as the original report.

In the case of 23-demethylgorgostane derivatives, C-22 and C-23 assignments are confusing around 24 ppm and 25 ppm signals. (Compounds 53~66, except compound 58.)

There are mysterious assignments for unique 23,24-didemethylgorgostane (58).

In comparison of 13C NMR data between those of compound 58 and those of compounds 57 and 59.

There are no signal around 16 ppm for C-29 of compounds 57 and 59 in the data for compound 58.

However, 13C signals for C-22~28: 25.1, 24.1, 45.0, 32., 18.1, 20.6, and 10.5ppm of compound 58 are quite similar to those of C-22~27 and C-29 of compounds 57 and 59. It is unclear by checking on the original report of ref. 37.

There would be many misassignments in addition to listed above.

Response:

Thank you very much for your comment, there is no mysterious assignment in compound 58, both 23-demethyl gorgostane and compound 58 have methylene carbon which appears around 10.5 ppm, this methylene carbon numbered (C-29) in 23-demethyl gorgostane but numbered (C-28) in 23,24-didemethylgorgostane as compound 58 due to the absence of terminal methyl (C-28) which appears around 16 ppm.

On structural drawing, there are many comments as listed below.

Figure 1 and Figure 2

Bold wedged bond between C17 and C20 is replaced by solid bond, and then add H-17 with hashed wedged bond.

Beta H-8, alpha H-9 and H-14 should be added.

Response:

Thank you very much for your comment, we modified (Figure 1 and Figure 2) as reviewer’s suggestion.

Figure 3

Bold wedged bond between C17 and C20 should be replaced by solid bond, and then add H-17 with hashed wedged bond. (Compds. 2,3,4,6, 9, 12, 13, 14, 15, 16, 17, 18, 19, 20, 22, 24, 25, 26, 27, 28, 29, 30, 31, 32, 33, 34, 36, 39, 42, 43, 45, 47, 49, 52, 54, 59, 60, 61, 62, 63, 64, 65, 66, 69, 70, 71, 74, and 75).

Response:

Thank you very much for your comment, we modified all these structures according to the reviewer’s suggestion.

Beta H-8, alpha H-9, H-14, and H-17 should be added except compounds possessing any functional group at C-8, C-9, C-14, and C-17. (All compounds).

Response:

Thank you very much for your comment, we modified all these structures as reviewer’s suggestion.

Many types of drawing are found for side chain portion. It should be unified.

Bold wedged bonds between C-22 and C-30, and C23 and C-30 on compound 1 would be recommended. (All compounds)

Bold wedged bond between C-23 and C-29 should be replaced by solid bond.

(Compds. 2, 3, 4, 9, 12, 13, 14, 15, 16, 18, 19, 20, 21, 22, 23, 24, 25, 26, 29, 31, 32, 35, 37, 38, 39, 40,41, 42, 43, 44, 47, 69, 71, and 75).

Hashed wedged bond between C-23 and C-29 should be also replaced by solid bond.

Response:

Thank you very much for your comment, we modified all these structures according to the reviewer’s suggestion.

- (Compds. 46, 48, 50, 72, and 73) H-22 and H-23 should be removed. (Compds. 38, 47, 48, 50, 53, 59).

Response:

Thank you very much for your comment, we modified H-22 and H-23 in (Compds. 46, 48, 50, 72, and 73) and (Compds. 38, 47, 48, 50, 53, 59) as reviewer’s suggestion.

There are C22 diastereoisomers, i.e. compounds 9, 14, and 18 (ref. 18).

Based on 13C NMR data around cyclopropane ring portion, it seems that stereochemistry on C22 would be same as to those of ordinary gorgostane compounds. If there are mistaken in drawing or assignments, please mention them in the text or as comments.

Response:

Thank you very much for your comment, we checked structural drawing in original report (Reference 18), the configuration of C-22 in compounds (9, 14, and 18) are (22S) configuration which different from the ordinary gorgostane. The determination of stereochemistry of these compounds in reference 18 (Jin et al., 2005) depending on NOESY NMR correlation (relative configuration not absolute configuration), moreover, they cited with (Tanaka et al., 2002) which assigned these carbons as (22R and 23R), this made conflict with the previous assignment, so, we modified these structures in manuscript according to absolute configuration determined by X-ray crystallography (Tanaka et al., 2002).

In relation to the stereochemistry on C-22 and C-23, please mention original work for the determination of absolute stereochemistry and using methods, such as X-ray and synthesis.

It is not only for gorgostane derivatives and also for 23-demethylgorgostanes.

Response:

Thank you very much for your comment, in gorgostanes the absolute configuration on C-22 and C23 were determined as (22R and 23R) by X-ray crystallography in (Tanaka et al., 2002). Moreover, the absolute configuration of naturally occurring 23-demethylgorgostanes is (22R and 23R) which determined by comparing naturally occurring 23-demethylgorgostanes with synthetic analogues. (Blanc, P. A., & Djerassi, C. 1980).

(Blanc, P. A., & Djerassi, C. (1980). Isolation and structure elucidation of 22 (S), 23 (S)-methylenecholesterol. Evidence for direct bioalkylation of 22-dehydrocholesterol. Journal of the American Chemical Society, 102(23), 7113-7114.)

There are many mistakes on compound name which should be revised.

Response:

Thank you very much for your comment, we modified these names according to the reviewer’s suggestion.

In addition to other minor comments, listed below.

From, L32 unique nuclei => core ring system

L41 Stoloniferone S => stoloniferone S

L44 Klyflac- => klyflac-……….etc,

To, I- Gorgost-5-ene: => I. Gorgost-5-ene:

II- 5,6-Epoxy gorgostane: => II. 5,6-Epoxygorgostane

III- 5,6-Dihydroxy gorgostane: => III. 5,6-Dihydroxygorgostane:

IV- 9,11-Seco-gorgostane: => IV. 9,11-Secogorgostane:

V- 23-Demethyl gorgostane: V. 23-Demethylgorgostane:

VI- => VI.

L.251 (1-75) => (1-75)

Response:

Thank you very much for your comment, all these minor comments have been covered in the manuscript, as reviewer’s suggestion.

Round 2

Reviewer 1 Report

Revised manuscript “Chemical review of Gorgostane type steroids isolated from marine organisms and their 13C-NMR spectroscopic data characteristics” by Abdelkarem and co-workers was checked. The reviewer still requires reconstruction of Figures and Tables. Why don’t the authors divide compounds by classification along Table 1? Only put them by numbers of compounds. For figures 15, for tables 8. Please imagine you are readers. Geometric relation between text, figures, tables is important for easy understanding for readers.

Author Response

Dear Editor and Reviewers,

Manuscript ID: marinedrugs-1578220

Type of manuscript: Review

Title: Chemical review of Gorgostane type steroids isolated from marine organisms and their 13C-NMR spectroscopic data characteristics

Thank you very much for your fruitful comments and recommendations, the requested comments are already included and highlighted in the main manuscript, here, the responds to reviewer comments.

Reviewer 1:

Comments and suggestions for Authors:

The reviewer still requires reconstruction of Figures and Tables. Why don’t the authors divide compounds by classification along Table 1? Only put them by numbers of compounds. For figures 15, for tables 8. Please imagine you are readers. Geometric relation between text, figures, and tables is important for easy understanding for readers.

Response:

Thank you very much. We reconstructed Tables and Figures, according to reviewer suggestions.

Reviewer 3 Report

Please see an attached file.

Author Response

Dear Editor and Reviewers,

Manuscript ID: marinedrugs-1578220

Type of manuscript: Review

Title: Chemical review of Gorgostane type steroids isolated from marine organisms and their 13C-NMR spectroscopic data characteristics

Thank you very much for your fruitful comments and recommendations, the requested comments are already included and highlighted in the main manuscript, here, the responds to each reviewer comments separately.

Reviewer 3:

Such misassignments will decrease the value of this article. Reviewer strongly recommends to revise the missassignments as mentioned and add comments in the footnote

Response:

Thank you very much for your recommendations. We revised the misassignments in the compounds and mentioned in the footnote that "those assignments are different from original report and reassigned in this report by careful comparison".

It’s impossible that 23-demethylgorgostane and 23, 24-demethylgorgostane have almost chemical shift value to the side chain portion within gamma position from C-24.

So reviewer recommend to modification on compound 58 as below

  1. Delete the structure and data of compound 58 from this review, because there are some mistakes in the original report.
  2. Keep the structure and data of 58, but add comments, those NMR data seems not supporting original structure.

Response:

Thank you very much for your recommendation, we kept the structure and data. Also we added a comment "those NMR data seems not supporting original structure” as reviewer recommend and after careful check.

There are still many mistakes in compound name which should be revised. In addition to other minor comments listed below.

Response:

Thank you very much for your comment. We corrected all typographical errors and respond to minor comments.

References

Please use appropriate abbreviations for journal

Response:

Thank you very much for your comment. We used appropriate abbreviations for journal.

Round 3

Reviewer 1 Report

The reviewer confirmed Figures were reconstructed. But table are not. The reviewer supposes that tables in the present style are good for looking, but not divided along classification of type of steroids. Then, the reviewer feels sorry that it is hard for readers to find data which are wanted for comparison each other.  

Author Response

Dear Reviewer,

Manuscript ID: marinedrugs-1578220

Type of manuscript: Review

Title: Chemical review of Gorgostane type steroids isolated from marine organisms and their 13C-NMR spectroscopic data characteristics

Thank you very much for your fruitful comments and recommendations, the requested comments are already included and highlighted in the main manuscript, here, the responds to reviewer comments.

 Reviewer 1:

Comments and suggestions for Authors:

The reviewer confirmed Figures were reconstructed. But table are not. The reviewer supposes that tables in the present style are good for looking, but not divided along classification of type of steroids. Then, the reviewer feels sorry that it is hard for readers to find data which are wanted for comparison each other.  Response:

Thank you very much for your comments and suggestions, we reconstructed Figures and NMR Tables, while Table 1 is already divided according to gorgostane sub-classes.

Reviewer 3 Report

Revised version is improved, but there are some mistakes. Please see attached file in which comments are listed.

Author Response

Dear Reviewer,

Manuscript ID: marinedrugs-1578220

Type of manuscript: Review

Title: Chemical review of Gorgostane type steroids isolated from marine organisms and their 13C-NMR spectroscopic data characteristics

Thank you very much for your fruitful comments and recommendations, the requested comments are already included and highlighted in the main manuscript, here, the responds to each reviewer comments separately.

Reviewer 3:

Comments and Suggestions for Authors:

All reviewer-3 comments and suggestions.

Response:

Thank you very much for your comments, we corrected all typographical errors, Tables, Figures, and references, according to reviewer suggestions.
